# Subtotal Pleurectomy with Intrathoracic Chemo Hyperthermia (HITHOC) for IVa Thymomas: De Novo Versus Recurrent Pleural Disease

**DOI:** 10.3390/cancers14205035

**Published:** 2022-10-14

**Authors:** Benjamin Chappuy, Gabrielle Drevet, Hugo Clermidy, Pascal Rosamel, Mickael Duruisseaux, Sebastien Couraud, Renaud Grima, Valentin Soldea, Lara Chalabreysse, François Tronc, Nicolas Girard, Jean-Michel Maury

**Affiliations:** 1Department of Thoracic Surgery, Lung and Heart-Lung Transplantation, University Hospital Louis Pradel, GH-Est, 69677 Lyon, France; 2Department of Anesthesiology and Critical Care Medicine, University Hospital Louis Pradel, GH-Est, 69677 Lyon, France; 3Department of Thoracic Oncology, University Hospital Louis Pradel, GH-Est, 69677 Lyon, France; 4Claude Bernard University Lyon 1 (UCBL1), 69677 Lyon, France; 5Department of Thoracic Oncology, University Hospital Lyon-Sud, GH-Sud, 69677 Lyon, France; 6Department of Pathology, University Hospital Louis Pradel, GH-Est, 69677 Lyon, France; 7Department of Thoracic Oncology, Institut Curie, 75005 Paris, France; 8RYTHMIC Network, Réseau Tumeurs Thymiques et Cancer, Institut Gustave Roussy, 114 rue Edouard Vaillant, 94805 Villejuif, France

**Keywords:** IVa Masaoka, thymomas, pleural involvement, thymic surgery, HITHOC, subtotal pleurectomy, intrathoracic chemotherapy

## Abstract

**Simple Summary:**

Stage IVa thymomas are rare entities in thoracic oncology without a standard of care. Evidence-based guidelines for the management of located pleural carcinomatosis are lacking. Surgery when feasible has an excellent prognostic factor. However, the technical choice is vast, from extra pleural pneumonectomy with high rates of morbidity and/or mortality to debulking with high rates of relapse. We investigated parietal subtotal pleurectomy combined with HITHOC in highly selected patients. The goal was to determine overall survival (OS) and the disease-free interval (DFI). Our findings suggest a real impact of this surgical procedure in distant relapse (DR) or de novo tumors (DNT). In this orphan disease, prospective and randomized trials are needed to clarify the place of HITHOC in the multimodal modern care of these patients.

**Abstract:**

Introduction: Stage IVa thymoma is a rare disease without a standard of care. Subtotal pleurectomy and HITHOC introduced in highly selected patients may provide interesting oncologic results. The purpose of this study was to distinguish de novo stage IVa tumors (DNT) from distant relapse (DR) with respect to post-operative and long-term outcomes to provide the procedure efficacy. Methods: From July 1997–December 2021, 40 patients with IVa pleural involvement were retrospectively analyzed. The surgical procedure was subtotal pleurectomy and HITHOC (cisplatin 50 mg/m^2^, mitomycin 25 mg/m^2^, 42 °C, 90 min). The post-operative outcome, disease-free interval (DFI) and overall survival (OS) were analyzed. Results: Mean age was 52 ± 12 years. B2 and B3 thymomas were preponderant (27; 67.5%). The median number of pleural nodes were nine (4–81) vs. five (1–36); *p* = 0.004 * in DNT and DR, respectively. Hospital mortality rate was 2.5%. There were four specific HITHOC complications (10%). DFI were 49 and 85 months (*p* = 0.02 *), OS were 94 and 118 months (NS), in DNT and DR, respectively. Conclusions: Subtotal pleurectomy with HITHOC in IVa offers satisfying results in highly selected patients, for both DNT and DR. Due to the disease rarity, multicentric studies are needed to define HITHOC as a standard of care.

## 1. Introduction

Thymomas arising from thymic epithelial cells are the most frequent anterior mediastinal tumors belonging to orphan thoracic oncology [1] with a French incidence of about 250–350 cases per year [2]. The association of tumoral thymic epithelial cells and normal lymphocytes respecting the normal architecture of the thymus characterizes thymomas. In most cases, thymomas present indolent behavior with a tendency of loco-regional invasion. However, in 5% of cases, thymomas can spread along serous membranes as pleura (70%) and pericardium, rather than lymph nodes or distant organs [3]. When feasible, surgical en bloc resection R_0_ is one of the most significant prognosis factors in the multimodal assessment therapy of thymomas [4,5]. In more detail, if surgery is performed in the early stages, surgery can be associated with postoperative radiotherapy following ITMIG recommendations, whereas neo-adjuvant chemotherapy is used for voluminous tumors with reasonable doubts of feasibility of initial R_0_ surgery [6]. Finally, the association of chemotherapy and radiotherapy is the standard association in unresectable tumors. Masaoka’s stage IVa is characterized as a tumor with microscopically confirmed separate nodes from the primary tumors involving the pleura (parietal and/or visceral) or pericardial and/or epicardium surface [7]. Stage IVa is generally identified during oncological workup by means of an Iodine-CT-Scan and/or 18-FDG-Pet-Scan, but can be accidentally discovered during thymic surgery. The curation of IVa includes multimodal strategies and innovations. Multimodal strategies have been purposed as combined chemo-radiations or surgery. However, the exact role of surgery is still unclear and controversial. Surgical options range from debulking (removal of 90% of entire tumors) to localized pleural surgery until extra-pleural pneumonectomy (EPP). The particularity of stage IVa is that it is a metastatic disease where R_0_ surgery is not feasible. Secondary to a controversial debulking surgery or the high rate of mortality/complications of EPP, some authors in the 1990’s advocated for a new concept derived from Sugarbaker experience for peritoneal/pleural malignant mesothelioma [8], introducing pleurectomy/decortication combined with intrathoracic chemo hyperthermia to cure stage IVa thymomas. The concept is based on a two-step surgery: (1) Subtotal pleurectomy associated with wedges and phrenic muscle resection where all visible tumor is removed, but in essence, the same as R_1_ and (2) A hyperthermic intracavitary cisplatin-based chemotherapy to destroy the microscopic tumor residue [9,10]. Due to the rarity of the disease, only a small number of specialized surgical teams can present retrospective studies evaluating the place of HITHOC in stage IVa thymomas. In highly selected patients, HITHOC may provide better results in progression-free survival and overall survival; however, in a rare oncologic disease, the difficulty lies in the identification of the sub-group to enhance results. In this setting, based on our experience in HITHOC, we retrospectively analyzed our results and introduced a comparison between the following two sub-groups: IVa de novo (DNT) and IVa distant relapse (DR) pleural spread.

## 2. Patients and Methods

### 2.1. Study Design

From July 1997 to December 2021, we retrospectively analyzed the prospectively collected data of patients with pleural involvement of thymomas who underwent cytoreductive subtotal pleurectomy followed by intrathoracic chemo hyperthermia (HITHOC). The study was conducted in the Department of Thoracic Surgery, Lung and Heart-Lung Transplantation, Louis Pradel Hospital, a university hospital of the Hospices civils de Lyon, and a French expert center for thymic malignancies labialized by the national RYTHMIC network (thymic tumors and cancer). Data were acquired from a prospective database and follow-up was collected from secured medical charts (Easily^®^ medical software, Hospices Civils de Lyon, France). According to French laws, the study protocol was approved and registered by the ethical committee of the SFCTCV (French society of thoracic and cardiovascular surgery, IRB: 00012919; number 2022-04-06-19470).

### 2.2. Aims of the Study

The primary end-point of the current study was to evaluate, in a large series of a rare tumoral disease, the results of HITHOC. The secondary end-point was to determine the role of oncological outcomes on de novo IVa (DNT) vs. distant relapse (DR), with special attention focused on disease-free interval (DFI) and overall survival (OS).

### 2.3. Preoperative Evaluation

All the patients underwent standard preoperative assessment for major thoracic surgery with complete preoperative history and thymic oncological history and physical examination. Morphological and oncological evaluation were realized with standard chest X-ray, iodine CT-Scan, 18-FDG-Pet-Scan. MRI was performed when there was suspected major vessel tumoral invasion. A specific cardiac evaluation was performed with echocardiography and stress test when there were major cardiovascular risk factors such as previous chemotherapy with CAP and mediastinal radiotherapy. Respiratory function tests were assessed. Finally, a complete blood count, liver function test, and renal function tests (including creatinine clearance tests) were conducted. From 1997 to 2012, according to French guidelines, all patients were presented to an institutional medical board including surgeons, onco-pneumologists, and radiotherapists. From 2012 to the present, the national board of the RYTHMIC network reviewed all patients. Inclusion criteria are defined elsewhere [11].

### 2.4. Surgical Technique and Devices

The protocol for pre-hydration, initially assessed by a mean of 2–3 L of saline serum, was stopped in 2018 and replaced by a specific protocol of nephro-protection by means of natrium thiosulfate in analogy of cisplatin nephro-protection used in intra-peritoneal chemo hyperthermia [12,13]. Natrium thiosulfate was intravenously released just before endocavitary chemotherapy at 9 mg/m^2^ over 20 min, then a new injection at 12 mg/m^2^ was instilled for 6 h immediately after surgery. All patients with myasthenia gravis (MG) were prepared for surgery with the intravenous perfusion of human immunoglobulin (Tegelin^®^, LFB Biomedicaments, France) to prevent a post-surgical acute MG crisis.

The HITHOC procedure was conducted under general anesthesia after placement of a thoracic epidural catheter. Venous and arterial monitoring was continuously assessed using internal jugular central and radial artery catheterization. Internal temperature was controlled using a bladder thermic catheter. All patients were positioned in the lateral decubitus position. A 5th or 6th posterolateral thoracotomy without serratus preservation was assessed (to avoid detachment spaces), and an initial extra-pleural dissection was performed. A subtotal parietal and phrenic pleurectomy was performed and lung or associated phrenic muscle resection were conducted when necessary [14] to achieved complete macroscopic tumor removal. After subtotal pleurectomy, parietal argon beam electrocoagulation was performed. Two 24 F-chest tubes were inserted and the thoracotomy was hermetically closed. The tubes were finally connected to the HITHOC device. The patient was re-positioned into a dorsal decubitus position and endocavitary chemotherapy was performed.

The HITHOC device was initially made of a local designed material (Cavitherm, Soframedical, Vienne, France), replaced in 2018 by the Sunship2 (Gamida, Eaubonne, France). Antimitotic agents were cisplatin 50 mg/m^2^ and mitomycin 25 mg/m^2^ perfused endocavitary at 42 °C with 1.4 L/m^2^ of isotonic saline solution over 90 min at 600–800 mL/min, according to central venous tension (<12 mmHg). This chemotherapy regimen was initially established following previous experiences in malignant pleural mesothelioma. At the end of the procedure, patients were transferred to the intensive care unit for tube removal and cardiothoracic monitoring.

### 2.5. Follow-Up

A complete follow-up was conducted using clinical and morphological data. Patients were routinely seen at 1, 3, and 6 months after discharge. Routine yearly CT scans were performed. Adverse events (AEs) were recorded using the National Cancer Institute Grading. Hospital mortality and morbidity were recorded within a <30 days post-surgery interval. Follow-up was obtained for all cases and censored on 31 December 2021.

### 2.6. Statistical Analysis

Statistical analysis was performed using GraphPad Prism (GraphPad) software. All tests were done using a significant threshold of α = 0.05. Overall survival (OS) and disease-free interval (DFI) were defined from the date of HITHOC and last follow-up of death or evidence of disease recurrence within the study period. OS and DFI were estimated using Kaplan–Meier’s method. Group comparisons were made using with the *t*-test or Mann and Whitney test. Multivariate analysis was unable to be performed due to the small sample size.

## 3. Results

### 3.1. Study Population

We performed cytoreductive subtotal pleurectomy with HITHOC on 40 patients with thymoma pleural involvement. The mean age was 52 ± 12 years, ranged 21–72, and there were 25 women and 15 men. Seventeen patients presented with myasthenia gravis (42.5%). Histopathologic findings according to WHO classification were: A (*n* = 2; 5%); AB (*n* = 1; 2.5%); B1 (*n* = 3; 7.5%); B1B2 (*n* = 2; 5%); B2 (*n* = 18; 45%); B2B3 (*n* = 5; 12.5%); and B3 (*n* = 9; 22.5%). There were 13 DNT and 27 DR. Main characteristics are presented in Table 1.

### 3.2. Intra-Operative Results

Of the total population studied (*n* = 40), 24 patients (60%) underwent associated lung wedge resections and 24 (60%) underwent partial phrenic muscle resections with direct non-resorbable stitches reinforced with Teflon^®^ patches. The median number of resected nodes was five, with a range of 1–81. The median number of resected nodes with significantly higher in the DNT group (9 [4–81] vs. 5 [1–36]; *p* = 0.004). Per-operative surgical resection results are shown in Table 2.

Four patients (10%) underwent extra-pleural pneumonectomy with HITHOC procedure due to previous surgical pleural symphyses. Seventeen (42.5%) patients had redux pleural surgery due to associated lobectomy (*n* = 3; 7.5%), superior vena cava replacement (*n* = 3; 7.5%), and wedge resections to upper lobes (*n* = 11; 27.5%) during thymic radical removal; moreover, eight (20%) patients underwent previous partial pleural resections for pleural relapse.

### 3.3. Post-Operative Outcomes

Post-operative complications were reported in 16 patients (40%). Nonspecific HITHOC complications included infectious pneumopathy (*n* = 6; 15%), prolonged air leak in two cases (5%) and two cases of pyothorax (5%). Specific HITHOC complications were reported as follow: four cases of (10%) acute renal insufficiency without extra renal epuration with spontaneous recuperation after rehydration. One patient experienced spontaneously reversible bone marrow insufficiency. Specific chemotherapy complications were no longer observed due to the natrium thiosulfate nephro-protection protocol. A significant statistical difference was observed between thiosulfate- vs. thiosulfate+ groups in terms of maximal plasmatic peak within the 7-day post-operative period, shown in Figure 1.

The median length of stay in the intensive care unit (ICU) and in the hospital were respectively: 1 day, ranged 1–26 and 10 days, ranged 6–36 days. The median length of chest tube removal was 5 days, ranged 4–32.

In-hospital mortality was 2.5%. One patient died at day 7 due to a multi-organ failure induced by septic post-operative pleuro-pneumonia. One patient who underwent an extra-pleural pneumonectomy died at 3 months due to multi-complicated bronchial fistulae; another patient one died at one year due to septic pneumonia.

### 3.4. Long-Term Outcomes

The median DFI after HITHOC was 70 months, with a significant difference between DNT and DR, being 49 vs. 85 months (*; *p* = 0.029), respectively. Eighteen patients (20%) presented subsequent disease recurrence. The most encountered sites of relapse were homolateral pleura in ten patients (25%), contralateral pleura in two patients (5%), and mediastinum in two patients (5%). One patient presented a liver metastasis (2.5%) and another in the axillary lymph node (2.5%); one patient presented lung metastasis (2.5%) and another presented multiple systemic relapses including the pericardium, lung, and spine. More details on the DFI of homolateral pleural vs. contralateral or systemic relapse is shown in Table 3.

The median length of survival after HITHOC was 118 months, without significant differences between DNT vs. DR. The median lengths of OS were respectively 94 vs. 118 months. Fifteen patients died during the entire study period, four (26%) due to septic shock induced by pneumonia, four (26%) due to disease progression, two (13.3%) from acute MG crisis, two (13.3%) due to heart failure induced by anthracyclin chemotherapy, one (6.6%) due to a paraneoplastic nephrotic syndrome, one due to COVID-19-induced ARDS, and one from an unknown cause. Survival data are shown in Figure 2, Table 3.

We analyzed the effect of the number of resected pleural nodules based on an artificially created threshold of 10. DFI was significantly different in patients with more than 10 resected nodules; medians were respectively 49 vs. 70 months; *p* = 0.043. This difference did not impact OS, with a median of 118 months in <10 pleural nodules group; median was not found for >10 Pleural nodules group. 

## 4. Discussion

Complete surgical resection is the cornerstone of the treatment of thymomas. Stage IVa with pleural and/or pericardial spread is a metastatic disease for which complete resection has been deemed unrealistic and intrinsically at least R_1_ or R_2_. Facing the difficulties in treatment of a metastatic but localized disease with slow evolution, HITHOC was introduced as a better alternative to extra-pleural pneumonectomy in terms of morbidity, mortality, and overall survival. It also delayed the need for systemic therapies that were used when treatment had limited efficacy in advanced disease [6,15].

In this study, we show that HITHOC is a safe surgical procedure in a trained multidisciplinary team, with an acceptable 30-day mortality rate of 2.5% and 90-day mortality of 5%, in light of the historical results of EPP (respectively 17 and 30% [16,17,18]. Meanwhile, the morbidity rate is mostly related to post-operative complications; interestingly, we were able to reduce the risk of acute renal failure by 10%. The natrium thiosulfate protocol demonstrates a positive influence on renal function, and should be a standard for the HITHOC procedure. We were able to increase cisplatine dosage to 100 mg/m^2^ without adverse effect, and may encourage increasing the dosage to 150–200 mg/m^2^. However, the optimal regimen for intrathoracic chemotherapy is still under question. The cisplatin regimen has been the most evaluated with positive results on tissue penetration, but there is a lack of rationale for a second drug [19,20]. In the present series, we used mitomycin, which has replaced doxorubicin for being more efficient in thymomas [21]. Unfortunately, given the rarity of TETs with pleural spread, a clinical evaluation of different chemotherapy regimens is challenging. In our opinion, a preliminary in vitro model using tumoral thymic epithelial cells should be used as a preliminary evaluation for other potential drugs [22].

DNT should not be a contra-indication and must encourage multi-modal treatment. This involves neo-adjuvant chemotherapy to reduce the volume of the thymic tumor, if not resectable (as recommended in the RYTHMIC guidelines) and a two-step surgery: (1) Radical thymectomy and, one or two months later, (2) Subtotal pleurectomy and HITHOC. The choice of a two-step surgery was guided by two compelling reasons: (1) The necessity of a median sternotomy for thymic tumor removal and (2) The use of a postero-lateral approach for HITHOC with a surgery duration of 6–7 h in the operating theatre. Nevertheless, DNT appears ro be a more aggressive disease with a higher rate of B2–B3 thymomas and a higher rate of pleural nodules adversely influencing DFI compared with DT. We report that the number of metastatic pleural nodules influence DFI without impact on OS. The presence of >10 pleural nodules was a significant factor of lower DFI and affected the extent of pleural invasion. Low et al. and Qayyum et al. [23,24] advocated for a MRI with higher sensitivity to predict the peritoneal carcinomatosis index (PCI) before the cytoreductive peritoneal procedure. Standard pre-operative surfaced MRI and PCI adapted to pleura would be helpful for the decision and prognosis of pleural cytoreductive HITHOC procedures. Moreover, in our experience, the number of metastatic pleural nodules encountered during surgery is always more important than those revealed by CT or 18-FDG scan and is, in our opinion, a major reason to prefer subtotal pleurectomy to tumorectomy in an R_1_ surgery.

One of the big questions currently unanswered is the veracity of HITHOC’s procedure. Several response elements can be added. Firstly, in the context of thymoma, it has been clearly demonstrated that the most complete surgical resection is a first-order predictive factor. Secondly, we were able to demonstrate using the French database RYTHMIC that surgical excision of pleural nodules (including HITHOC cases) in stage IVa compared to systemic treatments reduced the risk of recurrence by 60% (HR = 0.4, 95 CI (0.25–062); *p* < 0.0001) [25]. These results were confirmed by the study of Aprile et al. [26], who were able to demonstrate in a retrospective study that the HITHOC procedure coupled with a cytoreductive pleurectomy gave better results in DFI compared with cytoreduction alone: 88 ± 15 vs. 57 ± 19 months (*p* = 0.046). Moreover, our results are fully comparable with those previously published showing a 5-year survival rate between 70–100% and a DFI of 61–87% [27]. However, it remains difficult to interpret DFI alternatively published in the form of percent or months; in our opinion, DFI expression in months seems to be more informative for the oncologist. Finally, in the context of this disease, it remains very difficult to set up a randomized comparative study, which could answer these questions with greater certainty.

The evaluation of the HITHOC procedure in thymomas remains difficult due to the rarity of the disease. This leads to relative ignorance of standard of care and innovations by physicians. There is great heterogeneity in stage Iva thymoma, in terms of pleural lesion number, size, location, and time interval after initial treatment or diagnosis, which also hampers such assessments. We insist on the major role played by dedicated multidisciplinary networks such as RYTHMIC, ITMIG, ESTS, JART, and CHART to collect data and provide international standards of care. In our opinion, the evaluation of the real impact of HITHOC procedures should regroup all expert centers to define a multicentric study protocol. Evaluation should compare subtotal pleurectomy alone vs. subtotal pleurectomy with HITHOC vs. systemic therapies. The ultimate goal should be to define, in the era of personalized medicine, the best therapeutic option in selected patients to chronicize a metastatic disease with a good quality of life and achieve prolonged outcomes.

## 5. Conclusions

Subtotal parietal and phrenic pleurectomy combined with HITHOC seems to be a valuable option in Masaoka IVa thymomas in highly selected patients. We highlighted a median DFI of 49 and 85 in DNT and DR and OS of 94 and 118 months, respectively. This result offers the possibilities of a long therapeutic time-out in a metastatic disease with a high risk of relapse.

In our opinion, such cases must be discussed in dedicated multidisciplinary boards to improve and clarify result of HITHOC. The gold standard would be a randomized trial, which cannot be realized, in an orphan disease with indolent progression. International collaboration would be helpful to organize a prospective database in this setting.

## Figures and Tables

**Figure 1 cancers-14-05035-f001:**
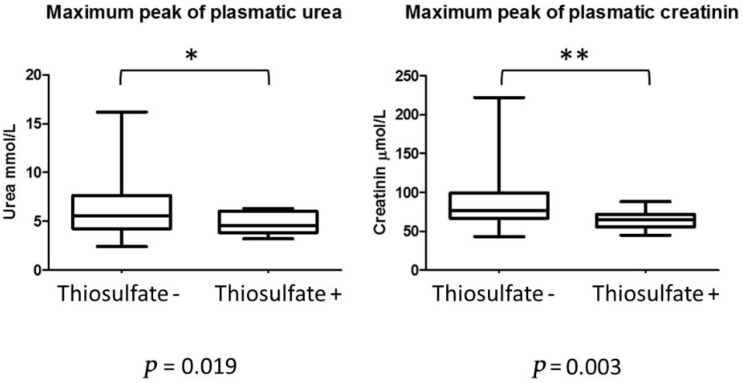
Post-operative peak of plasmatic levels of urea (µmol/L) and creatinine (mmol/L). * *p* < 0.05; ** *p* < 0.001.

**Figure 2 cancers-14-05035-f002:**
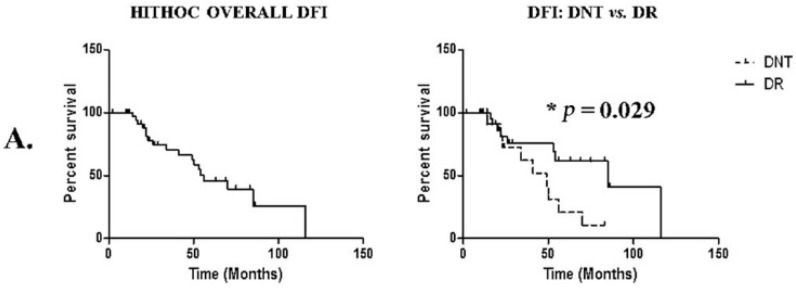
Disease-free interval (DFI) and overall survival (OS) overview. (**A**) Analysis of disease-free interval (DFI) of the entire cohort, comparing IVa de novo vs. distant relapse. (**B**) Analysis of overall survival (OS). (**C**) Analysis of DFI and OS in <10 or >10 resected pleural nodule groups.

**Table 1 cancers-14-05035-t001:** Population main characteristics.

	DNT	DR	Total
N	13	27	40
M/F	6/7	9/18	15/25
Age at HITHOC	51 ± 12; (21–65)	52 ± 12; (29–72)	52 ± 12; (21–72)
Age at Thymectomy	51 ± 12; (21–65)	46 ± 12.5; (28–70)	47 ± 13; (21–70)
Myasthenia Gravis N; %	6; 46%	11; 41%	17; 42%
Histopathology N; %			
A	/	2;7.5%	2;5%
AB	/	1; 3.7%	1; 2.5%
B1	2; 15.5%	1; 3.7%	3; 7.5%
B1/B2	/	2; 7.5%	2; 5%
B2	6; 40%	12; 44.5%	18; 45%
B2/B3	/	5; 18.5%	5; 12.5%
B3	5; 38.5%	4; 15%	9; 22.5%

**Table 2 cancers-14-05035-t002:** Intra-operative findings.

	DNT	DR	Total	*p*-Value
N	13	27	40	
Associated resection				
Lung wedges	9	15	24	
Phrenic muscle	8	16	24	
Resected nodes	22 ± 23; (4–81) *	7 ± 7; (1–36)	12 ± 15; (1–81)	*p* = 0.004
median	9	5	5	
1 < *n* < 10	7	21	28	
10 < *n* < 20	1	4	5	
20 < *n* < 30	0	5	5	
>30	5	1	6	

* the attention was focalized on the necessity of lung resection, and phrenic muscle resection. The number of resected nodes was evaluated during the pathologic analysis. The number of resected nodes was statistically higher in the DNT group.

**Table 3 cancers-14-05035-t003:** Oncologic outcomes, DFI, and OS.

	DNT	DR	Total
N=	13	27	40
FDI (median) months	49 *	85	70
OS (median) months	94	118	118
OS			
1 year	100%	88%	92%
5 years	100%	80%	86%
10 years	33%	49%	40%

* *p* < 0.05.

## Data Availability

The data can be shared up on request.

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
