# Peer review of "Subtotal Pleurectomy with Intrathoracic Chemo Hyperthermia (HITHOC) for IVa Thymomas: De Novo Versus Recurrent Pleural Disease"

_cancers, 2022, doi:10.3390/cancers14205035_

Round 1

Reviewer 1 Report

 Authors reported results of retrospective analyses of subtotal pleurectomy with HITHOC in patients with stage IVa thymoma. They concluded that subtotal pleurectomy with HITHOC in patients with stage IVa offers satisfying results in highly selected patients, both for de novo stage IVa tumors and recurrent tumors. This report is valuable for publication in Cancers. Therefore, this report should be allowed to be published after minor revision.

Major points

1. Mitomycin

 Mitomycin is a very classic drug and the authors should provide a clear scientific rationale for using it as part of the regimen. The authors should also suggest a next regimen that fits the modern era and looks to the future of thymoma treatment.

2. Techniques of closure for massive air leakage after decortication

 In this cohort, patients received subtotal pleurectomy, which might be equal pleurectomy and decortication. In this procedure, the aggressive decortication for the visceral pleura is performed, therefore massive air leakage should be observed. Unless a careful and persistent closure of the leakage is performed intraoperatively, postoperative complications, especially prolonged air leakage complicated with empyema, cannot be prevented. Furthermore, without closure of the air leakage, a large amount of harmful anticancer drugs would be aspirated into the pulmonary parenchyma, resulting in complications such as severe chemical pneumonitis. Therefore, authors should add more detailed information on the method and practice of closure for massive air leakage after decortication. Authors also provide the movie or video of those closure techniques of air leakage.

Author Response

Response to reviewer 1.

  1. Mitomycin

 Mitomycin is a very classic drug and the authors should provide a clear scientific rationale for using it as part of the regimen. The authors should also suggest a next regimen that fits the modern era and looks to the future of thymoma treatment.

Response: thanks for the remark. The use of mitomycin was a declination of our previous experiences in malignant pleural mesothelioma with HITHOC. I totally agree with you that only cisplatin has been studied in this setting and the rational for a second drug is questionable. In our experience, the use of mitomycin has been stopped after this 40 patients and we now use doxorubicin with a better rational as recommended in systemic therapies by the rythmic board.

As you suggest we add un the method section: “Cisplatin 50 mg/m2 and Mitomycin 25 mg/m2 endocavitary perfused at 42°C with 1,4 l/m2 of isotonic saline solution during 90 minutes at 600 – 800 ml/min according to central venous tension (< 12 mmHg), this chemotherapy regimen was initially instituted following previous experiences in malignant pleural mesothelioma)”

A comparative study between mitomycin and doxorubicin will be published later (I hope…).

This fact was discussed here “ Cisplatin regimen is the most evaluated with interesting results on tissue penetration but what is the rationale for a second drug 19,20 .In the present series, we used mitomycin finally replaced for doxorubicin more efficient in thymomas 21.”

  1. Techniques of closure for massive air leakage after decortication

 In this cohort, patients received subtotal pleurectomy, which might be equal pleurectomy and decortication. In this procedure, the aggressive decortication for the visceral pleura is performed, therefore massive air leakage should be observed. Unless a careful and persistent closure of the leakage is performed intraoperatively, postoperative complications, especially prolonged air leakage complicated with empyema, cannot be prevented. Furthermore, without closure of the air leakage, a large amount of harmful anticancer drugs would be aspirated into the pulmonary parenchyma, resulting in complications such as severe chemical pneumonitis. Therefore, authors should add more detailed information on the method and practice of closure for massive air leakage after decortication. Authors also provide the movie or video of those closure techniques of air leakage.

Response: thanks for this remark. I agree with you, my purpose was unclear. We performed a parietal and phrenic subtotal pleurectomy with associated localized nodules resection of the visceral pleura as described in our previous publication in annals of thoracic surgery. Lung parenchyma closure was realized using standard technics of thoracic surgery such as stiches or wedges. The chemo was instilled in the chest cavity on a non-ventilated lung to reduce this risk. We never experienced chemical pneumonitis as the other teams involved in HITHOC.

We add in the method section :” A subtotal parietal and phrenic pleurectomy was performed and lung or associated phrenic muscle resection were realized if necessary 14 to achieved a complete macroscopic tumoral removal.” We deliberately didn’t describe the technical tips to close air leaks secondary to the use of standard treatment used in thoracic surgery

Reviewer 2 Report

The purpose of the study was to evaluate the de novo stage IVa tumors (DNT) and recurrent tumors (RT) with respect to post-operative and long-term outcomes. 40 patients with IVa pleural involvement were retrospectively analyzed. The surgical procedure was subtotal pleurectomy and HITHOC (Cisplatin 50 mg/m2, Mitomycin 25 mg/m2, 42°C, 90 min.  Hospital, mortality was 2.5%. Specific HITHOC complications were 4 (10%). DFI were 49 and 85 months (p=0.02*), OS were 94 and 118 months (NS) in DNT and DR respectively. Authors claim that subtotal pleurectomy with HITHOC in IVa offers satisfying results in highly selected patient, both for DNT and RT.

1.     In abstract last sentence of Results “DR” please correct. RT or DR, same for Table 1.Sometimes used as DT. Please use single term.

2.     Please use the term pleural nodules rather than nodes (May cause confusion with mediastinal lymph nodes.

3.     How can the authors claim about satisfactory outcomes when they do not have data of those  who do not have HITHOC in the same stage.

4.     Please indicate in methods that “In metachronous group, DFS and OS starts from the pleural surgery not from the primary surgery”.

5.     In part discussion Authors claim that “DNT should not be a contra indication and must encourage a two steps surgery (1): radical thymectomy and one or two months later (2): subtotal pleurectomy and HITHOC. Nevertheless, DNT appears as a more aggressive disease with a higher rate of B2 – B3 thymomas with a higher rate of pleural nodes adversely influencing DFI compared to DT.”

6.     Please discuss  why the authors feel comfortable about delaying a  systemic chemotherapy  or do not perform a complete surgery in one setting including the HITHOC. Difficult to understand for a surgeon.

7.     Authors also claim that “We report that the number of metastatic nodes influence DFI without impact on OS. The presence of >10 pleural nodes seems to be a significant factor of lower DFI and lead to  how to assess the extent of pleural invasion during the surgical work out?” Especially after this finding, Do they recommend any other modality to these patients?

8.     Please include manuscripts on stage 4A disease without HITHOC and their outcomes especially with those published after 2015. After discussing the outcomes of 4A disease in different experiences you can support your findings.

9.     Another Group can think that the better survival obtained in these patients might be due to adjuvant oncologic treatment variabilities developed recently. If you look at the literature without HITHOC survivals look much better in the recently published series.

10.  The most important question is: What was the postoperative regimen and what oncology drugs were used after DFI is over to provide the long OS.  Please also commnet on localized radiotherapy applications if you had any after recurrences or to mediastinum if there is an extensive disease in the mediastinum.

Author Response

Response to reviewer 2.

  1. In abstract last sentence of Results “DR” please correct. RT or DR, same for Table 1.Sometimes used as DT. Please use single term.

Response: RT or DR has been changed in DR (Distant Relapse) as recommended.

  1. Please use the term pleural nodules rather than nodes (May cause confusion with mediastinal lymph nodes.

Response: Pleural nodes has been replaced by Pleural nodules as recommended.

  1. How can the authors claim about satisfactory outcomes when they do not have data of those  who do not have HITHOC in the same stage.

Response: dear reviewer, thanks for this important comment; hithoc surgery remains controversial and was purposed in highly selected patients. It is one of the limit of the study, however we demonstrated that pleural surgery) including hithoc decrease of 60ù the risk of relapse, we included this answer in the discussion section

“ One of the big questions currently unanswered is the veracity of HITHOC's procedure. Several response elements can be added. Firstly, in the context of thymoma, it has been clearly demonstrated that the most complete surgical excision possible is a first-order predictive factor.

secondly, we were able to demonstrate using the French database rythmic that surgical excision of pleural nodules in stages IVa compared to systemic treatments reduces the risk of recurrence by 60% (HR=0.4, 95CI (0.25 -062); p<0.0001). ref

Finally, in the context of a rare disease with a rather slow evolution, it remains very difficult and complicated to set up a randomized comparative study, which could answer with certainty.”

Maury, J.M. et al. Outcomes of thymic epithelial tumors (TETs) with pleural metastases: Real-world insight from RYTHMIC. J Clin Oncol 15, doi:10.1200/JCO.2021.39.15_suppl.8578 (2021).

  1. Please indicate in methods that “In metachronous group, DFS and OS starts from the pleural surgery not from the primary surgery”.

Response: this point was already presented in the methods section with : “ . Overall Survival (OS) and Free Disease Interval (DFI) were defined from the date of HITHOC and the last follow up of death and evidence of disease recurrence within the study period

  1. In part discussion Authors claim that “DNT should not be a contra indication and must encourage a two steps surgery (1): radical thymectomy and one or two months later (2): subtotal pleurectomy and HITHOC. Nevertheless, DNT appears as a more aggressive disease with a higher rate of B2 – B3 thymomas with a higher rate of pleural nodes adversely influencing DFI compared to DT.”
  2. Please discuss  why the authors feel comfortable about delaying a  systemic chemotherapy  or do not perform a complete surgery in one setting including the HITHOC. Difficult to understand for a surgeon.
  3. Authors also claim that “We report that the number of metastatic nodes influence DFI without impact on OS. The presence of >10 pleural nodes seems to be a significant factor of lower DFI and lead to  how to assess the extent of pleural invasion during the surgical work out?” Especially after this finding, Do they recommend any other modality to these patients?

Response 6 7 : changes were made in the discussion section. ”Neo-adjuvant chemotherapy to reduce the volume of thymic tumor if not resectable in first intention (as recommended in the RYTHMIC guidelines) and two steps surgery (1): radical thymectomy and one or two months later (2): subtotal pleurectomy and HITHOC. The choice of a two-step surgery was guided by two compelling reasons (1) the necessity of median sternotomy for thymic tumor removal and (2) the use of a postero-lateral approach for HITHOC with a surgery duration (6-7 hours in operating theatre)”

  1. Please include manuscripts on stage 4A disease without HITHOC and their outcomes especially with those published after 2015. After discussing the outcomes of 4A disease in different experiences you can support your findings.

Response: I understand that it would be perfect to compare hithoc to other cares. However, as mentioned in the methods section, patient were highly selected to undergo this procedure and are not comparable to non-surgical treatment. We are working to identify similar patient in our database to make a “cas-temoin” study in another paper.

  1. Another Group can think that the better survival obtained in these patients might be due to adjuvant oncologic treatment variabilities developed recently. If you look at the literature without HITHOC survivals look much better in the recently published series.

Response: as we said in discussion, the goal was to provide a long period without systemic treatment that will be required later in most cases. I do not agree with your comment.  Publications report a 30% rate response with systemic therapies. New therapies in metastatic thymomas are immune checkpoint therapies unfortunately non-indicated in patients with B1 or with paraneoplastic manifestations as myasthenia gravis or good syndrome. The use of rapamycin is complicated secondary to adverse effects encountered. The place of radiation therapies is indicated well-delimited lesions with a poor control. The only fact that can be discuss it that surgery in most publications offers better survival in DFI and OS and the place of hithoc is difficult to evaluate.

  1. The most important question is: What was the postoperative regimen and what oncology drugs were used after DFI is over to provide the long OS.  Please also comment on localized radiotherapy applications if you had any after recurrences or to mediastinum if there is an extensive disease in the mediastinum.

Response: in the multimodal care of such patients, in presence of relapse after hithoc, all patient are discussed within the RYTHMIC national board to define the introduction of systemic therapies or radiations according to the RYTHMIC guidelines. Your remark is an excellent remark to discuss the care modalities in such patients but it was not the goal of our study

thanks for your reviewing work

best regards